# Optimizing quantum gates towards the scale of logical qubits

Paul V. Klimov [1] ✉, Andreas Bengtsson [1], Chris Quintana[1], Alexandre Bourassa [1], Sabrina Hong [1], Andrew Dunsworth[1], Kevin J. Satzinger [1], William P. Livingston [1], Volodymyr Sivak[1], Murphy Yuezhen Niu[1], Trond I. Andersen[1], Yaxing Zhang[1], Desmond Chik[1], Zijun Chen[1], Charles Neill [1], Catherine Erickson[1], Alejandro Grajales Dau[1], Anthony Megrant [1], Pedram Roushan [1], Alexander N. Korotkov[1,2], Julian Kelly [1], Vadim Smelyanskiy[1], Yu Chen[1] & Hartmut Neven [1]

A foundational assumption of quantum error correction theory is that quantum gates can be scaled to large processors without exceeding the error-threshold for fault tolerance. Two major challenges that could become fundamental roadblocks are manufacturing high-performance quantum hardware and engineering a control system that can reach its performance limits. The control challenge of scaling quantum gates from small to large processors without degrading performance often maps to non-convex, high-constraint, and time-dynamic control optimization over an exponentially expanding configuration space. Here we report on a control optimization strategy that can scalably overcome the complexity of such problems. We demonstrate it by choreographing the frequency trajectories of 68 frequency-tunable superconducting qubits to execute single- and two-qubit gates while mitigating computational errors. When combined with a comprehensive model of physical errors across our processor, the strategy suppresses physical error rates by ~3.7× compared with the case of no optimization. Furthermore, it is projected to achieve a similar performance advantage on a distance-23 surface code logical qubit with 1057 physical qubits. Our control optimization strategy solves a generic scaling challenge in a way that can be adapted to a variety of quantum operations, algorithms, and computing architectures.

Superconducting quantum processors have demonstrated elements of surface code quantum error correction[1–6] establishing themselves as promising candidates for fault-tolerant quantum computing. Nonetheless, imperfections in hardware and control introduce physical errors that corrupt quantum information[7] and could limit scalability. Even if a large enough quantum processor with a high enough performance limit to implement error correction can be manufactured, there is no guarantee that a control strategy will be able to reach that limit.

Frequency-tunable architectures[4,8–19] are uniquely positioned to mitigate computational errors since most physical error mechanisms are frequency dependent[20–31] (Fig. 1a–d). However, to leverage this architectural feature, qubit frequency trajectories must be choreographed over quantum algorithms to simultaneously execute quantum operations while mitigating errors.

Choreographing frequency trajectories is a complex optimization problem due to engineered and parasitic interactions among

[1]Google AI, Mountain View, CA, USA. [2]Department of Electrical and Computer Engineering, University of California, Riverside, CA, USA. ✉e-mail: pklimov@google.com

computational elements[27] and their environment[22,26,29], hardware[32] and control[28] inhomogeneities, performance fluctuations[26,33], and competition between error mechanisms. Mathematically, the problem is non-convex, highly constrained, time-dynamic, and expands exponentially with processor size.

Past research into overcoming these complexities employed frequency partitioning strategies that either faced difficulties scaling with realistic hardware imperfections[34,35] or whose scalability is not well understood[4,13,17,36]. To overcome the limitations of these strategies, we proposed the Snake optimizer[37,38] and employed an early version in past reports[39–48]. However, an optimization strategy has not been developed around it, it has not been rigorously benchmarked, and large enough processors to investigate its scalability have only recently become available. Whether high-performance configurations exist at scale and whether they can be quickly discovered and stabilized are open questions.

Here we address these questions by developing a control optimization strategy around Snake that can scalably overcome the complexity of problems like frequency optimization within the high performance, high stability, and low runtime requirements of an industrial system. The strategy introduces generic frameworks for building processor-scale optimization models, training them for various quantum algorithms, and adapting to their unique optimization landscapes via Snake. This flexible approach can be applied to a variety of quantum operations, algorithms, and architectures. We believe it will be an important element in scaling quantum control and realizing commercially valuable quantum computations.

We investigate the prospects of this strategy for optimizing quantum gates for error correction in superconducting qubits. We demonstrate that it strongly suppresses physical error rates, approaching the surface code threshold for fault tolerance on our processor with tens of qubits. To pave the way towards much larger processors, we demonstrate Snake "healing" and "stitching", which were designed to stabilize performance over long timescales and geometrically parallelize optimization. Finally, we introduce a simulation environment that emulates our quantum computing stack and combine it with optimization, healing, and stitching to project the scalability of our strategy towards thousands of qubits.

## Results

### Quantum hardware

Our hardware platform is a Sycamore processor[6] with $N = 68$ frequency-tunable transmon qubits on a two-dimensional lattice. Engineered tunable coupling exists between 109 nearest-neighbors[49,50]. We configure our control system and processor to execute the surface code gate set, which includes single-qubit XY rotations (SQ) and two-qubit controlled-Z (CZ) gates[10] (Supplementary Note 1). SQ gates are implemented via microwave pulses resonant with qubits' respective $|0\rangle \leftrightarrow |1\rangle$ idle frequencies ($f_i$ for qubit $q_i$ executing SQ$_i$). CZ gates are implemented by sweeping neighboring qubits into $|11\rangle \leftrightarrow |02\rangle$ resonance near respective interaction frequencies ($f_{ij}$ for qubits $q_i$ and $q_j$ executing CZ$_{ij}$) and actuating their couplers. The $N = 68$ idle and ~$2N = 109$ interaction frequencies—which constitute one frequency configuration $F$ with dimension ~$3N = |F| = 177$—parameterize qubit frequency trajectories, which we seek to optimize.

### Performance benchmark

We evaluate the performance of frequency configurations via the parallel two-qubit cross-entropy benchmarking algorithm (CZXEB, see Supplementary Note 6)[39,51]. CZXEB executes cycles of parallel SQ gates followed by parallel CZ gates, benchmarking them in a context representative of many quantum algorithms. Most relevant to this study is that CZXEB reflects the structure of the surface code's parity checks and has empirically served as a valuable performance proxy of logical error[6]. The processed output of CZXEB is the benchmark

distribution $e_c$, in which each value is one qubit pair's average error per cycle $e_{c,ij}$, which includes error contributions from respective SQ$_i$, SQ$_j$, and CZ$_{ij}$ gates. Benchmarks are generally not normally distributed across a processor and are thus reported via percentiles as $50.0\%^{(97.5-50)\%}_{(2.5-50)\%}$ and plotted as quantile boxplots. The wide range from 2.5% to 97.5% is the distribution spread, which spans $\pm 2\sigma$ standard deviations for normally distributed data.

### Optimization model

We approach frequency optimization as a model-based problem. In turn, we must define an algorithm error estimator $E$ that is representative of the performance of the target quantum algorithm $A$ at the optimizable frequency configuration $F$ (Fig. 1d). This problem is hard because the estimator must be fast for scalability, predictive for scaling projections, and physical for metrology investigations. We introduce a flexible framework for overcoming these competing requirements that can be adapted to define the optimization landscapes of a variety of quantum operations, algorithms, and architectures.

Our framework corresponds to the decomposition $E(F|A, D) = \sum_{g \in A} \sum_{m \in M} w_{g,m}(A) \epsilon_{g,m}(F_{g,m}|D)$ where the sums are over all gates $g \in A$ and known physical error mechanisms $m \in M$. $\epsilon_{g,m}$ are algorithm-independent error components that depend on some subset of frequencies $F_{g,m} \subseteq F$ and can be computed from relevant characterization data $D$ (Supplementary Note 2 and 3). $w_{g,m}$ are algorithm-dependent weights that capture algorithmic context via training on benchmarks that are sufficiently representative of $A$. Defining the estimator thus maps to defining the target quantum algorithm, the algorithm-independent error components, and then training the algorithm-dependent weights.

We set our target quantum algorithm to CZXEB to gear the estimator towards the surface code's parity checks. Furthermore, since CZXEB is also our benchmarking algorithm, we can associate the performance of optimized frequency configurations with our optimization strategy. We then define error components corresponding to dephasing[23–25], relaxation[23,26,30,31], stray coupling[27], and frequency-pulse distortion[28] over qubit frequency trajectories. The relevant characterization data include qubit flux-sensitivity spectra, energy-relaxation rate spectra, parasitic stray coupling parameters, and pulse distortion parameters, which are measured prior to optimization. Finally, we train the weights via a protocol[52] that we developed specifically to reduce the risk of overfitting (Supplementary Note 4). It constrains weights via homogeneity and symmetry assumptions and then leverages the frequency tunability of our architecture to train them on single- and two-qubit gate benchmarks taken in configurations of variable complexity.

The resulting algorithm error estimator represents a comprehensive understanding of physical errors across our processor. It spans ~$4 \times 10^4$ error components, only has 16 trainable weights for the full processor, and is trained and tested on ~6500 benchmarks. Despite its scale, it can still be evaluated ~100 times/s on a desktop. Furthermore, it can predict CZXEB cycle errors in the wide range ~$3$–$40 \times 10^{-3}$ within two factors of experimental uncertainty (Supplementary Note 4). In total, the estimator fulfils our speed, predictivity, and physicality requirements.

### Optimization strategy

Finding an optimized frequency configuration from the algorithm error estimator maps to solving $F^* = \text{argmin}_F E$. This problem is hard for several reasons. First, all $|F| \sim 3N$ idle and interaction frequencies are interdependent due to engineered and parasitic interactions between nearest and next-nearest neighbor qubits. Second, the estimator has numerous local minima since most error mechanisms and hardware constraints compete, and since it is built from noisy characterization data. Finally, there are ~$k^{|F|} \sim k^{3N}$ possible configurations, where $k$ is the number of options per frequency, as constrained by hardware and control specifications and inhomogeneities. In total, the problem is

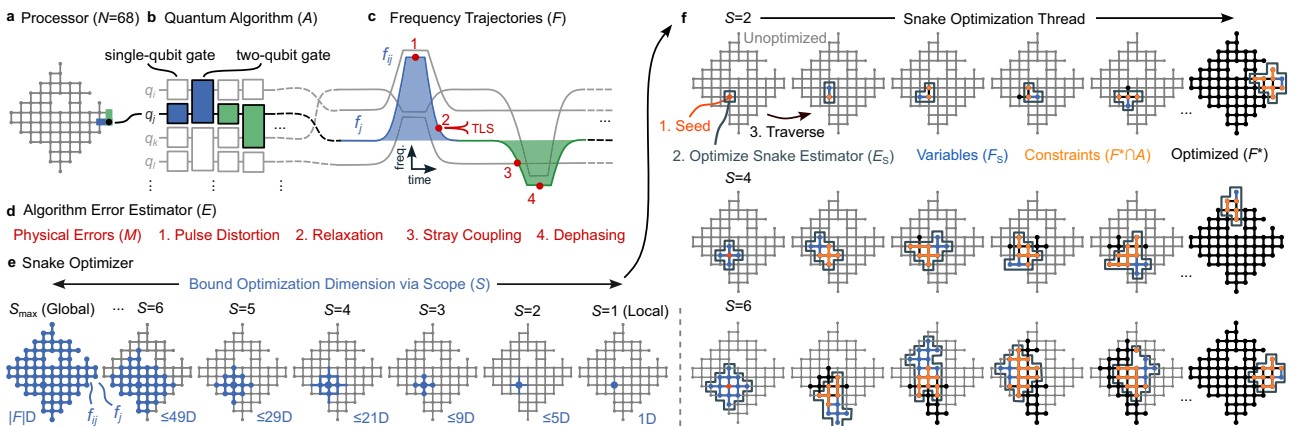

**Fig. 1 | Frequency optimization. a** Our quantum processor with $N = 68$ frequency-tunable superconducting transmon qubits represented as a graph. Nodes are qubits (e.g., black dot) and edges are engineered interactions between them (e.g., blue and green bars). **b** A quantum algorithm ($A$) comprising single- and two-qubit gates with one qubit ($q_j$) distinguished. **c** Corresponding qubit frequency trajectories ($F$), parameterized by single-qubit idle ($f_j$ for qubit $q_j$) and two-qubit interaction ($f_{ij}$ for $q_i$ and $q_j$) frequencies. Quantum computational errors depend strongly on frequency trajectories since most physical error mechanisms are frequency dependent (red dots are non-exhaustive examples). Namely, pulse distortion errors (1) increase with larger frequency excursions. Relaxation errors (2) increase near relaxation hotspots, for example due to two-level-system defects (TLS, horizontal resonance). Stray coupling errors (3) increase near frequency collisions between coupled computational elements. Dephasing errors (4) increase towards lower frequencies, where qubit flux-sensitivity grows. **d** We leverage our understanding of physical error mechanisms ($M$) to estimate the algorithm's error ($E$) and then optimize it with respect to qubit frequency trajectories. **e** We employ the Snake optimizer, which can solve optimization problems at an arbitrary dimension (D), controlled by the scope parameter ($S$). These graphs show possible idle (nodes) and interaction (edges) frequency optimization variables (blue) at one Snake optimization step for scopes ranging from $S = S_{max}$ (global limit, $|F|$D optimization) to $S = 1$ (local limit, 1D optimization). **f** Snake optimization threads (progress horizontally) for three scopes (increase downwards). Snake's high configurability enables it to scalably overcome frequency optimization complexity and be adapted to a variety of quantum operations, algorithms, and architectures.

highly-constrained, non-convex, and expands exponentially with processor size. We developed the Snake optimizer[38] to scalably overcome the complexity of control optimization problems like frequency optimization.

Snake implements a graph-based algorithm that maps the variable frequency configuration $F$ onto a graph and then launches an optimization thread from some seed frequency (Fig. 1e, f). It then finds all unoptimized frequencies $F_S$ within a neighborhood whose size is bounded by the scope parameter $S$, and constructs the Snake estimator $E_S$. $E_S$ contains all terms in $E$ that depend only on $F_S$, which serve as optimization variables, and previously optimized frequencies $F^*$ that are algorithmically relevant $F^* \cap A$, which serve as fixed constraints. Snake then solves $F_S^* = \mathrm{argmin}_{F_S} E_S$, updates $F^*$, traverses, and repeats until all frequencies have been optimized. Frequency configurations are typically optimized from multiple seeds in parallel and the one that minimizes the algorithm error estimator is benchmarked.

Snake's favorable scaling properties are derived from the scope $S$, which tunes the greediness of its optimization between the local and global limits[38]. By tuning the scope within $1 \le S \le S_{max}$, we can bound the number of frequencies optimized at each traversal step to $1 \le |F_S| \le |F|$, where $|F_S| \sim S^2$ and $S_{max} \sim \sqrt{3N}$ (Fig. 1e). In turn, we can split one complex ~$3N$-dimensional problem over ~$k^{3N}$ configurations into ~$3N/S^2$ simpler ~$S^2$-dimensional problems over $\sim k^{S^2}$ configurations each. Such splitting terminates at $S = 1$, where Snake optimizes ~$3N$ 1-dimensional problems over ~$k^1$ configurations each. Importantly, the intermediate dimensional problems with $S < S_{max}$ are exponentially smaller than the global problem and independent of processor size.

Snake is not expected to discover globally optimal configurations. However, if it can find sufficiently performant configurations for the target quantum algorithm – for example with errors below the fault-tolerance threshold[3]–it will solve the scaling complexity problem. Namely, we will not be faced with an exponentially expanding problem as our processors scale, but linearly more problems with bounded configuration spaces. Furthermore, since Snake's seed strategy, traversal strategy, inner-loop optimizer, and scope are highly configurable (Supplementary Note 5), it should be adaptable to overcome similar scaling complexities in other control problems and hardware.

## Validating performance

To experimentally investigate whether Snake can actually find performant frequency configurations at some intermediate dimension, we optimize our processor at scopes ranging from $S = 1$ (177 1D local problems) to $S = S_{max}$ (one 177D global problem) and benchmark CZXEB. We evaluate configurations by comparing their benchmarks against three performance standards (Fig. 2a). First, the baseline standard references benchmarks taken in a random frequency configuration, which establishes the average performance of the hardware and control system without frequency optimization ($e_c = 16.7^{+267.1}_{-10.6} \times 10^{-3}$ at $N = 68$). Second, the outlier standard references a constant cycle error, above which gates are considered performance outliers ($e_c = 15.0 \times 10^{-3}$ for all $N$). Third, the crossover standard references published benchmarks from the same processor that reached the surface code's crossover regime, which approaches the error correction threshold ($e_c = 6.2^{+7.6}_{-2.5} \times 10^{-3}$ at $N = 49$)[3,6]. This standard establishes what we consider high performance, while recognizing that much higher performance will be necessary to implement error correction in practice.

The wide performance gap between the baseline and crossover standards is closed via frequency optimization (Fig. 2b). Namely, intermediate dimensional optimization ($2 \le S \le 4$) approaches the crossover standard ($e_c = 7.2^{+19.9}_{-2.5} \times 10^{-3}$ in ~130 s at $S = 2$) while suppressing performance outliers, with < 10% of gates above the outlier standard and < 0.5% failing calibrations, which prevent benchmarking. However, local ($e_c = 9.8^{+231.8}_{-6.0} \times 10^{-3}$ in ~6 s at $S = 1$) and global ($e_c = 10.8^{+145.0}_{-5.7} \times 10^{-3}$ in ~6500 s at $S = S_{max}$) optimization only marginally outperform the baseline standard. The optimal scope is $S = 4$ (≤21D optimization), but we default to $S = 2$ (≤5D optimization), which offers a better balance between performance and runtime (Supplementary Note 5). Next, we interpret.

First, the fact that we see performance variations between configurations illustrates that poor frequency choices cannot be compensated for by other components of our control system and that optimization is critical. Second, the fact that local optimization underperforms illustrates that frequency optimization is a non-local problem and that tradeoffs between gates must be considered. Third, the fact that global optimization underperforms even after an hour of

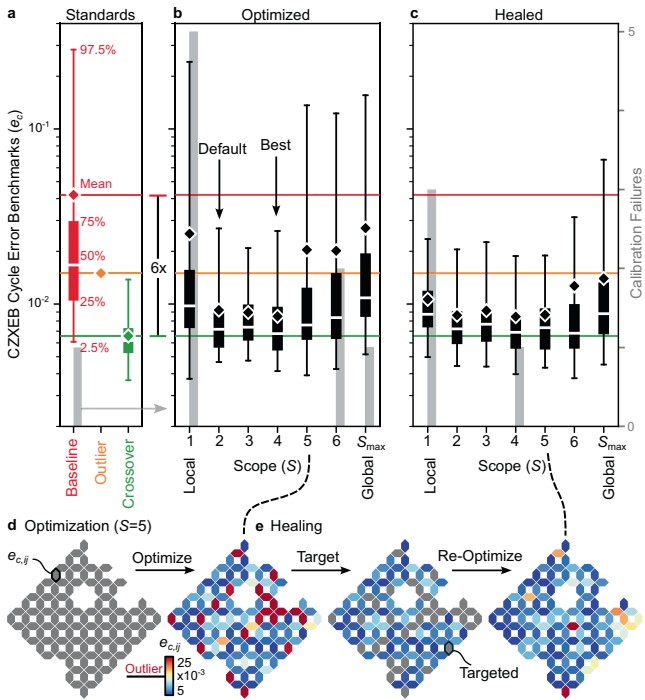

**Fig. 2 | Optimization and healing performance. a** CZXEB cycle error benchmarks ($e_c$, boxes, left axis) and calibration failures (gray bars, right axis in (**c**)) for the random baseline (red), outlier (orange diamond), and crossover (green) performance standards used to evaluate frequency configurations and our optimization strategy. Each box shows the 2.5, 25, 50, 75, and 97.5th percentiles and mean (see annotations on the baseline). The standards' means are extended across panels for comparison. **b** Benchmarks for configurations optimized at different scopes ($S$) ranging from $S = 1$ (local limit, 1D optimization) to $S = S_{max}$ (global limit, $|F|$D optimization). Intermediate dimensional optimization ($2 \leq S \leq 4$) outperforms both local and global optimization, finding configurations near the crossover standard. $S = 4$ ($\leq$21D optimization) performs best, with the lowest mean error, but $S = 2$ ($\leq$5D optimization) offers a better balance between performance and runtime, and is set as our default. **c** Benchmarks for each configuration in (**b**) after healing, which significantly suppresses performance outliers. Each box in (**a**), (**b**), and (**c**) corresponds to a distinct configuration. **d** Benchmark heatmaps illustrating optimization and (**e**) healing of targeted gates in the $S = 5$ ($\leq$29D optimization) configuration. Each hexagon corresponds to the cycle error for one pair ($e_{c,ij}$). Performant gates are blue, outliers are red, and unoptimized and targeted gates are gray.

searching illustrates the difficulty of navigating the configuration space even on our relatively small processor. Finally, the fact that relatively low intermediate dimensions found the most performant configurations is consistent with relatively local engineered and parasitic interactions and suggests that Snake can navigate our architecture's configuration space in a way that should scale to larger processors.

### Stabilizing performance

Stabilizing performant configurations is as difficult and important as finding them. Namely, a processor's optimization landscape constantly evolves and performance outliers emerge on timescales ranging from seconds to months, with the most catastrophic due to TLS defects fluctuating into the path of qubit frequency trajectories[26,33]. Unfortunately, even a low percentage of outliers can significantly degrade the performance of a quantum algorithm[6]. However, re-optimizing all gates of a processor when a low percentage of outliers are detected is unscalable from a runtime perspective and introduces the risk of degrading performant gates.

By design, Snake healing can surgically re-optimize outliers, nominally much faster than full re-optimization, and without degrading performant gates[38]. To investigate the viability of healing, we heal

all configurations generated by the variable-scope experiment described above (Fig. 2c–e), targeting poorly performing gates (Supplementary Note 8). From the perspective of stability, the progressively worse configurations emulate the performance of our processor over progressively longer timescales following optimization. Healing suppresses outliers by ~48% averaged over configurations, typically runs >10 × faster than full reoptimization, and rarely degrades performant gates. Furthermore, heals can be applied repetitively and parallelized for sufficiently sparse outliers. These results demonstrate the viability of healing for scalably suppressing outliers to stabilize performance.

### Impact of metrology

We now consider the impact of the algorithm error estimator's composition on Snake's performance. In particular, the dephasing, relaxation, stray coupling, and pulse distortion error components may be interpreted as distinct error mitigation strategies that can be activated independently. To isolate their impact and to understand their interplay, we progressively activate them in all combinations, optimize, and benchmark CZXEB (Fig. 3a).

To build intuition for the impact of each error mitigation strategy, we inspect frequency configurations optimized with only one mitigation strategy activated (Fig. 3b). Most are visually structured, with inhomogeneities arising from fabrication imperfections in the processor's parameters. Dephasing mitigation biases qubits towards their maximum frequencies, where flux sensitivity vanishes[9]. Relaxation mitigation biases qubits away from relaxation hotspots driven by coupling to the control[9] and readout circuitry[22,53], packaging environment[29], and random TLS defects[26]. Stray-coupling mitigation disperses qubits to avoid frequency collisions between parasitically coupled gates[27]. Finally, pulse-distortion mitigation biases idles towards a multi-layered checkerboard, with neighbors at one of two symmetric $|11\rangle \leftrightarrow |02\rangle$ CZ resonances[10], and interactions towards resonance between the idles, to minimize frequency excursions. The inversion of frequencies at the eastern edges of the processor was triggered by fabrication imperfections that broke the symmetry between CZ resonances. This observation highlights non-trivial interplay between error mitigation and hardware inhomogeneities.

Interestingly, while some of these mitigation strategies alone may find performant configurations at the scale of several qubits, none of them substantially outperform the random baseline configuration at the scale of our processor. As we progressively activate mitigation strategies, competition between error mechanisms causes frequency configurations lose visual structure, while performance approaches the crossover standard. Analyzing error contributions in optimized configurations, we confirm that activating mitigation strategies selectively and effectively suppresses their corresponding error components, while only weakly impacting others (Supplementary Note 8). These results support our interpretation of error components as error mitigation strategies and that our optimizer can effectively reconcile their competition and suppress them. More generally, they highlight the importance of error metrology on the performance of our optimization strategy.

### Performance scalability

We are finally ready to investigate Snake's scalability. To do so, we conduct a scaling experiment that may be valuable for evaluating the prospects of any quantum hardware and control system. Namely, we optimize, heal, and benchmark hundreds of configurations of our processor ranging in size from $N = 2$ to 68 (Fig. 4a). As before, we reference the crossover standard. However, we now reference multiple baseline standards that correspond to unoptimized random configurations of variable size. Despite the irregular shapes of some configurations, we find surprisingly clear scaling trends.

CZXEB benchmarks grow and then saturate in both optimized and unoptimized configurations. Furthermore, mean cycle errors are

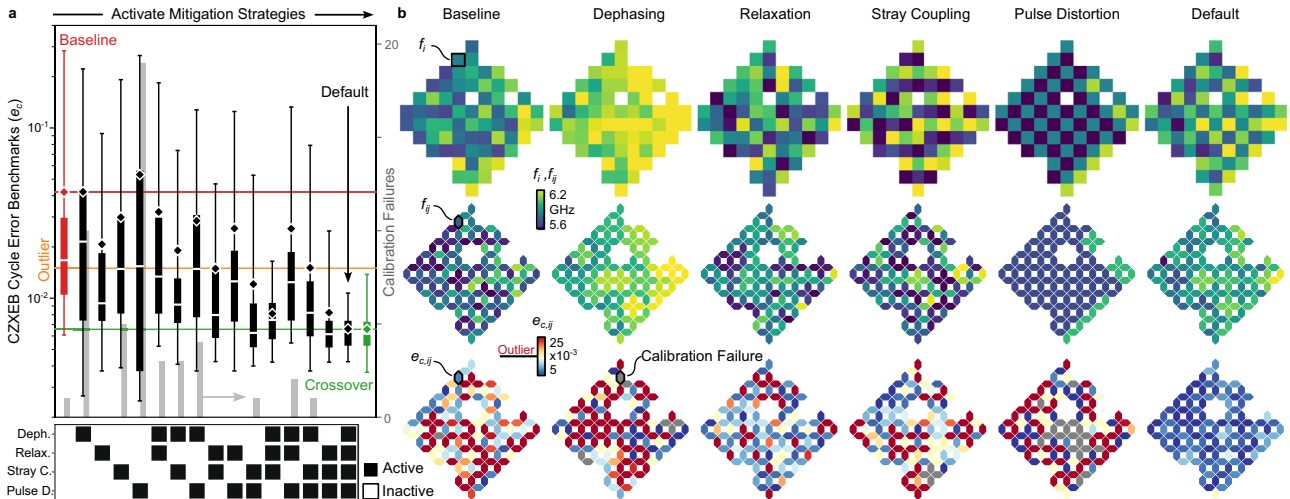

**Fig. 3 | Optimization performance versus error mitigation strategy. a** CZXEB cycle error benchmarks ($e_c$, black boxes, left axis) and calibration failures (gray bars, right axis) for configurations optimized with all combinations of dephasing, relaxation, stray coupling, and frequency-pulse distortion error mitigation strategies activated (see lower matrix). The random baseline (red), outlier (orange), and crossover (green) standards are shown and their means are extended across the panel for comparison. **b** Idle frequency ($f_i$, first row), interaction frequency ($f_{ij}$, second row), and cycle error ($e_{c,ij}$, third row) heatmaps for the baseline standard with no mitigation strategies activated (first column), configurations with only one strategy activated (central columns), and the default configuration with all strategies activated (last column). As more mitigation strategies are progressively activated (from left to right in (**a**)), cycle errors and calibration failures trend downwards, highlighting the importance of metrology on the performance of our optimization strategy.

well-represented by the model $\langle e_c(N) \rangle = e_{sat} - e_{scale} \exp(-N/N_{sat})$, where $N_{sat}$ is the qubit saturation constant, $e_{scale}$ is the error penalty in scaling gates from small to large systems, and $e_{sat}$ is the saturated error. Fitting this model to the empirical benchmarks, we find that optimized configurations saturate near the crossover standard, with best-fit parameters $N_{sat} = 22 \pm 10 (\pm 1\sigma)$, $e_{scale} = 3.1 \pm 0.4 \times 10^{-3}$, and $e_{sat} = 7.5 \pm 0.4 \times 10^{-3}$.

To estimate Snake's performance advantage, we make several comparisons. From the empirical benchmarks, we compare the mean cycle errors $\langle e_c^{base} \rangle / \langle e_c^{snake} \rangle$ in isolation ($N = 2$) and in parallel at scale ($N = 68$), which are $3.1 \pm 0.5$ and $6.4 \pm 1.0$, respectively. Remarkably, the optimized $N = 68$ configuration outperforms unoptimized $N = 2$ configurations by $2.3 \pm 0.4\times$. Furthermore, the optimized $N = 68$ configuration has a ~$40\times$ narrower benchmark distribution spread than the unoptimized $N = 68$ configuration. From the saturation model, we compare the scaling penalty $e_{scale}^{base} / e_{scale}^{snake}$ and the saturated cycle errors $e_{sat}^{base} / e_{sat}^{snake}$, which are $5.6 \pm 1.8$ and $3.7 \pm 0.7$, respectively. These comparisons illustrate that Snake achieves a significant performance advantage to $N = 68$.

To investigate Snake's future scalability, we simulate much larger processors than those manufactured to date. To do so, we developed a generative model that can generate simulated processors of arbitrary size and connectivity with simulated characterization data that are nearly indistinguishable from our processor[54]. We generate simulated processors ranging in size from $N = 17$ to $1057$, with connectivity corresponding to distance-3 to 23 surface code logical qubits ($d = 3$ to $23$ with $N = 2d^2 - 1$)[55]. We optimize simulated processors exactly like our processor and predict CZXEB benchmarks via our estimator (Supplementary Note 7). Simulated benchmarks reproduce the saturation trends seen in experiment, building trust in our simulation environment and results (Fig. 4b). Furthermore, they project that Snake's performance advantage should scale to a $d = 23$ logical qubit with $N = 1057$.

**Runtime scalability**

Despite the promising performance outlook, practically scaling to thousands of qubits will require Snake to be geometrically parallelized. Namely, even though optimization runtimes scale nearly linearly with processor size (~$3.6 \pm 0.1$ s added per qubit at $S = 2$), $N = 1057$ threads

take ~1.4 h. This exceeds our runtime budget of 0.5 h (Supplementary Note 5), which was chosen for compatibility with operating large surface codes.

By design, Snake stitching can split a processor into $R$ disjoint regions, optimize them in parallel, and stitch configurations[38]. Stitching leads optimization runtimes to scale sub-linearly with processor size, which should enable scalability towards $N \sim 10^4$ with $R = 128$ within our runtime budget in principle (Supplementary Note 5). In practice, however, stitching risks amplifying outliers at seams, where Snake must reconcile constraints between independently optimized configurations.

To investigate the viability of stitching, we stitch and heal our $N = 68$ processor with $R = 2$ (Fig. 4c) as well as an $N = 1057$ ($d = 23$) simulated processor with $R = 4$ (Fig. 4d). We chose convenient stitch geometries, but believe they will ultimately need to be optimized (Supplementary Note 8). Experimental data are limited, but outliers are not amplified at seams and stitched configurations perform as well as their unstitched counterparts ($e_c = 6.4^{+4.4}_{-1.8} \times 10^{-3}$ for $N = 68$ and $e_c = 6.3^{+4.3}_{-2.9} \times 10^{-3}$ for $N = 1057$). Finally, we note that stitching the $d = 23$ logical qubit with $R = 4$ is equivalent to stitching four $d = 11$ logical qubits into a 4-logical-qubit processor[55], which illustrates how larger surface codes may be optimized.

## Discussion

We introduced a control optimization strategy that combines generic frameworks for building, training, and navigating the optimization landscapes presented by a variety of quantum operations, algorithms, and architectures. It offers a significant performance advantage for quantum gates on our superconducting quantum processor with tens of qubits, approaching the surface code threshold for fault tolerance, and shows promise for scalability towards logical qubits with thousands of qubits. A recent demonstration of error suppression in a scaled-up surface code logical qubit[6] enabled by this strategy underscores its potential.

Elements of our strategy have also been employed to optimize quantum operations including measurement[56] and SWAP gates[40], and quantum algorithms for optimization[40], metrology[41,42], simulation[43–47], and beyond classical computation[39,48]. The strategy should also find value in quantum hardware beyond superconducting circuits[20],

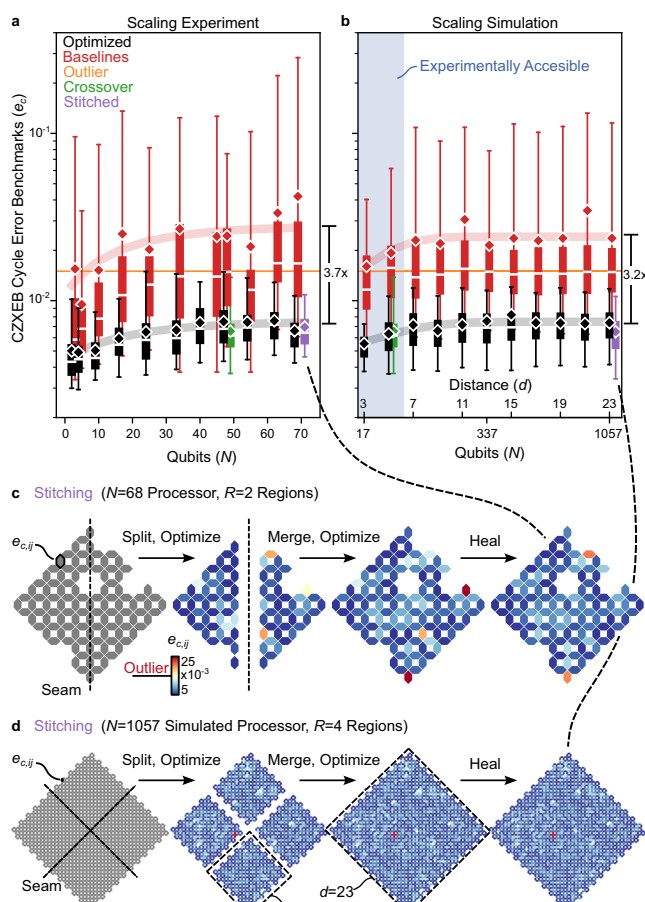

**Fig. 4 | Optimization scalability. a** Experimental and **b** simulated CZXEB cycle error benchmarks ($e_c$, boxes) in optimized (black) and unoptimized baseline (red) configurations of variable size. Simulated processors have size and connectivity corresponding to surface code logical qubits with distance $d$. The crossover standard (green), outlier standard (orange), and stitched configurations (purple) are shown for comparison. The solid lines are fits of the saturation model to the optimized (black) and baseline (red) benchmark means. Some boxes have been horizontally shifted to reduce overlap. In (**a**), $N < 40$ boxes combine benchmarks from multiple configurations to boost statistics. The x axis in (**b**) is linear in $d$ with $N = 2d^2 - 1$ and the shaded region illustrates the experimentally accessible regime of our processor. **c** Benchmark heatmaps illustrating stitching of our $N = 68$ processor and (**d**) $N = 1057$ simulated processor. Outliers are not substantially amplified at seams (dashed lines), which is our primary concern. The dashed regions in (**d**) illustrate that stitching the $d = 23$ logical qubit with $R = 4$ is equivalent to stitching four $d = 11$ logical qubits.

which face control challenges with similar scaling complexities. Choreographing the trajectories of electrons in quantum dots[57–59], shuttled ions in ion traps[60–62], or neutral atoms in reconfigurable atom arrays[63,64] are promising applications (Supplementary Note 3) that are of contemporary interest.

Looking towards commercially valuable quantum computations, significant challenges remain. The larger and more performant processors that are necessary to implement them will be susceptible to error mechanisms that are currently irrelevant or yet to be discovered. Furthermore, we expect that stabilizing performance over long computations that may span days[65] will present significant hurdles. Towards that end, Snake's model-based approach can leverage historical characterization data to forecast and optimize around failures before they happen[66,67]. Finally, even though we expect that model-based optimization will remain critical for injecting metrological discoveries into control optimization for the foreseeable future, Snake can also deploy model-free

reinforcement learning agents[68–70], which may reduce the burden of developing performance estimators (Supplementary Note 5). The techniques presented here should complement the numerous other control, hardware, and algorithm advancements necessary to realize commercial quantum applications.

## Data availability

The minimum dataset necessary to interpret, verify, and extend the research in this article is available in Supplementary Tables 1–5. Additional data are available from the corresponding author upon request.

## Code availability

The mathematical algorithm underlying the Snake optimizer and the pseudo code necessary to implement it are available in ref. 38.

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

## Acknowledgements

We thank the broader Google Quantum AI team for fabricating the processor, building and maintaining the cryogenic system, and general hardware and software infrastructure that enabled this experiment. We also thank Austin Fowler, Alexis Morvan, and Xiao Mi for their feedback on the manuscript.

## Author contributions

P.V.K. conceived, prototyped, and led the development of the Snake optimizer, algorithm error estimator, and simulated processor generative modeling frameworks. A.Be. and C.Q. contributed to engineering the

optimizer. A.Bo. and A.D. contributed to engineering the generative model. C.Q, A.Bo., A.D., K.J.S., M.Y.N., W.P.L., V.S., T.I.A., and Y.Z. contributed to error metrology research. S.H. led the development of parallel calibration infrastructure with engineering contributions from A.Be. and Z.C. D.C., C.N., C.E., and A.G.D. contributed to infrastructure. A.M., P.R., A.N.K., J.K, V.S., Y.C., and H.N. supported research and development.

## Competing interests

The authors declare no competing interests.
