## [Peer Review File · Nature Communications]

REVIEWER COMMENTS

Reviewer #1 (Remarks to the Author):

In the manuscript "Optimizing Quantum Gates Towards the Scale of Logical Qubits," Klimov and colleagues put forth a scalable optimization strategy for frequency configuration when implementing a specific quantum algorithm. This method involves the construction of an error estimator and the utilization of the optimizer to pinpoint a frequency configuration that minimizes errors. Through both experiment and numerical simulation, the authors showcase the efficacy and scalability of this optimization approach, recording a 3.7-fold enhancement compared to the baseline. Given the paramount importance of calibrating future large-scale quantum processors, the work undertaken here is commendable. While the manuscript is largely coherent and well-structured, I have concerns about its novelty and broader impact. The introduced SNAKE optimizer closely resembles those depicted in the authors' prior contributions. And while there's an emphasis on the error estimator, the manuscript falls short in elaborating on the noise model and the rationale behind this focus. Furthermore, the findings might be specifically tailored to their hardware, potentially limiting their broad applicability. This might narrow the paper's appeal to a wider audience.

More Comments:

1. Error Estimator Construction: The details provided in SI.V.B need more clarity. The authors mention, "The algorithm error estimator should include error components for all known physical error mechanisms," but this seems at odds with the aim of enhancing runtime efficiency. Adopting such an approach might inadvertently lead to overfitting when adjusting the weights. Figure 3 implies that cycle errors can be notably reduced compared to the baseline by activating specific error mitigation methods (e.g., Dephase, Relax, Stray C or Relax, Stray C, and Pulse Distortion). This points towards potential redundancies in the error estimator, possibly evidenced in the trained weights. Despite this, the authors choose not to present these weights, deeming them as not universally beneficial. However, revealing these weights might offer readers deeper insights into the error mechanisms, streamlining the error estimator process.

2. Optimization's Novelty: The manuscript delves deeply into the workflow and features of the SNAKE optimizer (including snake healing and snake stitching). Yet, much of this information appears to tread on familiar ground. The optimization segment seems to be a repurposing of an established technique in the domain, without proffering any substantial advancements or addressing potential challenges inherent to its application. A statement made by the authors, for instance, reads: "SNAKE is not anticipated to locate globally optimal configurations. Yet, if it identifies configurations that are sufficiently competent at any intermediary scale, it resolves the scaling complexity concern." It's

ambiguous where the boundaries of 'sufficient competence' lie, and whether it signifies a balancing act between performance and intricacy.

3. Metrology Investigation: The discourse on metrology in section VIII seems cursory. The authors elaborate on their methodology: "To discern their impact and comprehend their interactions, we sequentially activate them in varied combinations, optimize, and benchmark CZXEB." Nonetheless, the exposition appears restricted to frequency configurations with singular mitigation strategies in play. Such a methodology yields an oversimplified visual representation, sidestepping a nuanced analysis of the interactions among various mitigation approaches. The concluding remark in this segment, emphasizing the criticality of error metrology on the optimization strategy's performance, feels too generalized and lacks depth.

Reviewer #2 (Remarks to the Author):

The manuscript has a strong contribution to scalable quantum control; I'd recommend acceptance with minor revision. Here I detail my assessments by outlining the most important strengths and weaknesses.

Strengths:

* A software approach to scalably optimize the tunable qubit architectures is presented. As hardware size and capabilities grow, manual optimization will have a hard time keeping up. I believe the field will see software advances playing a significant role going beyond the NISQ era. And this work is a timely demonstration.

* The "snake optimizer" orchestrates the frequency configurations for running algorithms by a divide-and-conquer, model-based optimization strategy. A near linear (to system size) running time scaling is shown while achieving a significant performance advantage.

* Experimental validation is presented on quantum systems with ~70 qubits. Furthermore, simulated results are also shown to forecast applicability to future systems with ~1000 qubits (for tens of logical qubits).

* Presented evidence that software-based algorithm-specific optimization is critical.

Weaknesses:

Overall, the manuscript outlines a series of novel software techniques for improving the performance of hardware to match algorithms' demands. However, I believe there are some areas where the software evaluation could be strengthened. I'm optimistic that with the suggested improvements, the manuscript would tell readers a much stronger case about the significance of the snake optimizer.

* Limited benchmarks. The manuscript presented an extensive analysis of the cross-entropy/randomized benchmarking circuits, but unfortunately stops short of demonstrating error-correcting operations, such as syndrome measurements or logical operations. The title indicates that the optimization tool flow would contribute to scaling up to logical qubits, which got me super excited to see how the software could bring us closer to the demonstration of error-corrected logical qubits. In reality, the software is not tailored to error correction, but is general for any applications. I am sure readers would appreciate the work much better if the performance analysis included more benchmark circuits.

* Limited software details. The manuscripts omitted some details of the software (e.g., in appendices), making the software hard to reproduce. I suspect this is due to intellectual property considerations, as the software is not open-sourced. For this reason, as a reviewer, I cannot check the functionality of the tool or the reproducibility of the results. As a result, my assessment is primarily focusing on evaluation methodology and results.

* Some clarification questions:

1. The optimization relies on an accurate "algorithm error estimator". Can the authors elaborate on why a linear model (across different error mechanisms) sufficient?

2. The running time budget of the estimator and subsequently the overall tool flow is determined based on the impact of system drifts and outlier probability. How does this budget change as hardware evolves? Can we forecast quantitatively how fast the optimization needs to be for error correction?

3. Is the ellipsis in SI, section II there to indicate some parameters are omitted? For what reasons? Similarly, for the error components in SI, section V.B.

4. There is a claim in "Outlook" that the optimizer can deploy model-free RL agents. Do we expect a crossover point where model-based optimization will no longer be feasible and model-free models will be required? At the moment, it is hard to see how the model-based optimization techniques

proposed in this work will persist and carry over to scale beyond thousands of qubits. With that said, I completely agree with the authors that there's value in fleshing out what we can do at the 100-1000 qubit scale first.

Reviewer #3 (Remarks to the Author):

Dear Editor,

I have read the work "Optimizing quantum gates towards the scale of logical qubits", by Klimov et al., which the Google Quantum AI team present a classical strategy to optimize frequency controls of all qubits in a quantum processor, with the goal of minimizing the expected error per gate.

The work makes various interesting points, in my opinion. One is the algorithm itself, which is a greedy optimization method to select frequency schedules for qubits, so as to minimize the total expected error. The second one, equally important in my opinion, is the formulation of the cost function itself. This is a nontrivial step, because an inadequate error model leads to an inadequate optimization. Finally, the work itself is a tour-de-force calibration of a very large quantum processor to very good accuracy, for current standards, which makes this work a state-of-the-art reference --- absent from other large quantum computer providers--- and demonstrates the utility of the method.

However, there are presentation problems in the material, which is quite dense and at points does not offer information which is critical to understand the algorithm and its performance. Therefore, I would like the authors to address my comments below before committing to a final acceptance decision.

First of all, I think that error model information should not be completely relegated to the supplementary material, because it can be included with different wordings or a few sentences:

1. Section IV omits important facts regarding the error model. The first one, which was not obvious to me (and is still not fully clear) is that the error model only cares about which gates are present in the algorithm and not the order of execution or repetition.
2. After various readings of the manuscript and supplementary material, I still don't understand what structure is followed by the weights " $w_{\{g,m\}}$ " and how they are calibrated.

3. The third important fact which is absent from Sect. IV is that the formula at the end of page 2 assumes homogeneous parameters. This was implicit in the sentence "has 16 trainable parameters", but was unclear to me -- I originally thought 16 trainable parameters per error component.

4. The second paragraph on page 3 enumerates the error models. That list seems shorter than the one in the supplementary, which ends with some dots "... (?!?). It would not hurt splitting the number 4×10^4 as a product of error models times number of subsets, or a more informative list that is consistent with the procedure. This decomposition is also incomplete in the supplementary material.

I also have a fundamental issue with the choice of the words "frequency trajectory choreography", because an essential component of a choreography is the order in which steps and poses are executed. This ingredient seems to be absent in the error model and the optimization itself, unless I am mistaken, which is why I had such a difficult time understanding the formulation. If what I say is correct, the authors should correct this clearly in the abstract and the introductory text.

There are also presentation issues with the optimizer:

1. The optimization algorithm, while clever, seems like a standard greedy optimization method, which is the first thing one would try. This method seems to be explained in an e-print from 2006 (<https://arxiv.org/pdf/2006.04594.pdf>), which the authors properly cite, but it is unclear to me whether there has been any evolution. The original manuscript already contained the notions of scope, greedy search and stitching, but lacked a proper description of the error model / cost function and had no experimental data, which are the two big assets in this contribution. The authors should more clearly describe whether the optimization strategy is identical or new ingredients have been explored.

2. The authors should declare, even if tentatively or on average, the number of values "k" that frequencies can take. It is unclear from the supplementary material, whether the figure of 2^{1437} corresponds to the total configurations for one qubit, or the total processor.

3. Regarding the previous point, I am confused by the fact that the authors suggest a discretized frequency variable, but the Snake algorithm seems to rely on continuous optimization locally, according to III.3 in the supplementary material, from their mention of L-BFGS and similar algorithms.

4. It was also unclear to me whether the greedy optimizer goes over optimized values more than once. On a second read, it does not seem so, but this is not to be discarded in this type of algorithms. Some words on this would be helpful.

Other questions:

- The lack of finer-grained parallelization is attributed to the problem with stitching configurations. Wouldn't that be solved by allowing the healing process to run over optimized qubits?

- I find the analysis of the different error contributions quite interesting, though it is misleading the use of the word "error mitigation strategy", because the error mitigation may (or may not) result from that particular cost function. In this regard, I find the surface plot associated to pulse distortion quite puzzling, in that the optimization should find a uniform landscape, instead of strongly biasing two-qubit gates on the south-east boundaries. Given the homogeneous error parameters, this result and other inhomogeneities in the other surface plots merit discussion.

Response Key:

- Main revisions are **highlighted yellow** in the Main Text and Supplementary Information.
- Responses are labeled:
 - “**Response X.Y.**” with **X=Reviewer Number, Y=Question Number.**
 - Revisions use labels “**PX.LY.**” with **X=Page Number** and **L=Line Number.**

Reviewer #1 (Remarks to the Author):

In the manuscript "Optimizing Quantum Gates Towards the Scale of Logical Qubits," Klimov and colleagues put forth a scalable optimization strategy for frequency configuration when implementing a specific quantum algorithm. This method involves the construction of an error estimator and the utilization of the optimizer to pinpoint a frequency configuration that minimizes errors. Through both experiment and numerical simulation, the authors showcase the efficacy and scalability of this optimization approach, recording a 3.7-fold enhancement compared to the baseline. Given the paramount importance of calibrating future large-scale quantum processors, the work undertaken here is commendable. While the manuscript is largely coherent and well-structured, I have concerns about its novelty and broader impact. **The introduced SNAKE optimizer closely resembles those depicted in the authors' prior contributions.** And while there's an emphasis on the error estimator, **the manuscript falls short in elaborating on the noise model and the rationale behind this focus.** Furthermore, **the findings might be specifically tailored to their hardware, potentially limiting their broad applicability.** This might narrow the paper's appeal to a wider audience.

We thank the Reviewer for their time and effort in reviewing our manuscript and for their constructive comments, which we took significant steps to address. We believe they have helped us distinguish this work from our prior contributions, elaborate on new developments - including the noise model - and clarify the broad applicability of this work.

Response 1.1:

To address “ SNAKE optimizer closely resembles those depicted in the authors' prior contributions”:

We agree with the Reviewer that we could better distinguish this work from our original theoretical proposal for Snake (*Klimov, et al. arxiv:2006.04594 (2020)*) and highlight the significant technical advancements that were necessary to deliver it.

In this work, we introduce novel model construction and training frameworks and merge them with Snake into a holistic optimization strategy. We introduce concrete implementations for Snake's parameters for the first time, enabling us to explore tens of optimization strategies between the local and global optimization limit. Finally, we experimentally benchmark

optimization, healing, and stitching and introduce a simulation framework to project the scalability of our strategy to much larger processors than currently available.

Revisions:

1. Main Section I.

- **P1.L61. Added** that *“To overcome the limitations of [past] strategies, we proposed the flexible “Snake” optimizer (cited) and employed an early version in past reports (cited). However, an optimization strategy has not been formulated around it, it has not been rigorously benchmarked, and large enough processors to investigate its scalability have only recently become available.”*
- **P1.L74. Added** that *“The strategy introduces generic frameworks for building processor-scale optimization models, training them for various quantum algorithms, and adapting to their unique optimization landscapes via Snake.”*
- **P2.L3. Added** that *“To pave the way towards much larger processors, we demonstrate Snake “healing” and “stitching”, which were designed to stabilize performance over long timescales and geometrically parallelize optimization. Finally, we introduce a simulation environment that emulates our quantum computing stack and combine it with optimization, healing, and stitching to project the scalability of our strategy towards thousands of qubits.”*

2. Supplement Section V. Introduction.

- **P8.L52. Added** that *“In our past proposal, we established Snake's theoretical foundations and software abstractions. In this section, we map frequency optimization into Snake. We then provide broadly applicable implementations of its parameters and employ them to explore tens of optimization strategies between the local and global optimization limits over thousands of simulated optimization threads. Finally, we promote the most promising strategies to the experiments in the main text.”*

Response 1.2:

To address “the manuscript falls short in elaborating on the noise model and the rationale behind this focus”:

We agree that the noise model is an important component of this work and that additional details on its construction and the rationale behind it would be useful.

Revisions:

1. Supplement Section III Introduction:

- **P2.L29. Expanded** to say *“The algorithm error estimator can be represented by a variety of models (e.g. a linear model, a neural network, or a quantum simulation). The only strict requirement for optimization is that the estimator be representative of the performance of the target quantum algorithm. For this study,*

we append several requirements - physicality, speed, and accuracy. These are competing interests that are difficult to satisfy simultaneously. In the following sections we describe how we construct an estimator that can satisfy them.

2. Supplement Section III.A.:

- **P2.L39. Added** the rationale behind each assumption underlying the noise model and additional construction details.
- **P3.L8. Added** a list that describes how the model can be extended beyond our investigation and that we focused on a linear model because it “*establishes a powerful and broadly applicable framework:*”
 - *Compatible with physicality, speed, and accuracy.*
 - *Error components can be added as new error mechanisms are discovered.*
 - *Error components can be interchanged to transfer the estimator between hardware architectures (reference to new SI Section IIID).*
 - *Weights can be re-trained to transfer the estimator between quantum algorithms.*
 - *Optimization variables (F here) can be interchanged to adapt the estimator to other control variables, provided that the error components are defined in terms of those variables.*
 - *Equivalent to the basis-expansion method in machine learning (cited), bridging the domains and facilitating knowledge transfer.*
- **P3.L27. Added** that “*methods like randomized compiling (cited) offer the potential to recompile*” algorithms beyond our assumptions into our framework.

3. Supplement Section III.B.

- **P4.L16. Added** to the bulleted list many details on how our error components are segmented / defined.

Response 1.3:

To address “the findings might be specifically tailored to their hardware, potentially limiting their broad applicability”:

Our findings should transfer directly to various frequency-tunable superconducting qubit architectures, which are used across the world in academia (e.g. Berkeley, CalTech, Chalmers, ETH Zurich, KIT, MIT, Stanford, TU Delft, UChicago, USTC) and industry (e.g. Alibaba, Amazon, Rigetti, and some IBM research). Moreover, we believe they will apply to control problems in quantum dots, trapped ions, and neutral atoms. We took significant steps to highlight the broad applicability of our work to control problems in other hardware platforms.

Revisions:

3. Main Section V.

- **P3.L110. Clarified** that “*since Snake’s seed strategy, traversal strategy, inner-loop optimizer, and scope are highly configurable, it should be adaptable to*

overcome similar scaling complexities in other control problems and hardware architectures (see SI)", referencing **new Supplement Section III.D** (see below).

4. Main Section XI.

- **P7.L54. Added** that "The strategy should also find value in quantum computing architectures beyond superconducting qubits (cited), which face control challenges with similar scaling complexities. Choreographing the trajectories of electrons in quantum dots (cited), shuttled ions in ion traps (cited), or neutral atoms in reconfigurable atom arrays (cited) are promising applications (see SI) that are of significant contemporary interest", referencing **new Supplement Section III.D** (see below).

5. Supplement Section III.A.

- **P3.L13. Added** a list that describes how the estimator can be adapted to other control problems, including that "Error components can be interchanged to transfer the estimator between hardware architectures", referencing **new Supplement Section III.D** (see below).

6. Added new Supplement Section III.D. titled "Extensions to other hardware"

- **P4.L56. Added** a paragraph with 10 new references connecting our methods to other hardware, including " ... To illustrate the versatility of our framework, we connect our error components to control objectives when choreographing the spatial trajectories of reconfigurable atom arrays (cited). The objective of minimizing atom loss and heating by avoiding crossing spatial trajectories is analogous to our stray coupling components, which penalize crossing frequency trajectories. The objective of minimizing atom move distance is analogous to our pulse distortion components, which penalize large frequency excursions. The objective of minimizing atoms' vertical extent is similar to our dephasing components, which squeeze frequencies towards their maxima. Similar connections exist to control problems in other hardware, for example shuttling electrons in quantum dots (cited) or shuttling ions in ion traps (cited)."

7. Supplement Section V.C.:

- **P10.L46. Added** that Snake's seed, traversal, scope, and inner-loop optimizers must be tuned to the problem of interest and that "We interpret this tuning as adapting Snake to the error landscape presented by the algorithm error estimator and expect it to be especially critical when applying our strategy beyond our hardware."
- **P10.L60. Added** that "Once tuned, the parameters are remarkably robust. We have seen one set of tuned parameters remain reasonably effective for a variety of quantum algorithms and multiple generations of frequency tunable qubits, some of which underwent significant architectural modifications."

8. Supplement Section V.C.3.:

- **P11.L42. Added** that "we expect the optimal scope to depend strongly on the hardware architecture and to scale with the spatial extent of engineered and parasitic interactions. Optimizing hardware with higher connectivity (e.g. with three-qubit gates) would likely benefit from a larger scope than lower connectivity

(e.g. with two-qubit gates). Similarly, optimizing hardware with longer-range stray coupling would benefit from a larger scope than shorter-range stray coupling.”

More Comments:

1. Error Estimator Construction: The details provided in SI.V.B need more clarity. The authors mention, "The algorithm error **estimator should include error components for all known physical error mechanisms**," but this seems at odds with the aim of enhancing runtime efficiency. Adopting **such an approach might inadvertently lead to overfitting** when adjusting the weights. Figure 3 implies that cycle errors can be notably reduced compared to the baseline by activating specific error mitigation methods (e.g., Dephase, Relax, Stray C or Relax, Stray C, and Pulse Distortion). This **points towards potential redundancies** in the error estimator, possibly evidenced in the trained weights. Despite this, the authors choose not to present these weights, deeming them as not universally beneficial. However, **revealing these weights might offer readers deeper insights into the error mechanisms**, streamlining the error estimator process.

The Reviewer touches on several important details, which we segment and address below:

Response 1.4:

To address why the “estimator should include error components for all known physical error mechanisms”, “such an approach might inadvertently lead to overfitting”, and “points towards potential redundancies”:

We agree that our estimator construction and training protocols could use clarifications.

Revisions:

1. Main Section IV:

- **P3.L8. Added** that the estimator must include error components for all known error mechanisms because it must be *“physical for metrology investigations”*.
- **P3.L37. Added** that to combat overfitting, *“we train the weights via a protocol that we developed specifically to reduce the risk of overfitting (see SI). It constrains weights via homogeneity and symmetry assumptions and then leverages the frequency tunability of our architecture to train them} on single- and two-qubit gate benchmarks taken in configurations of variable complexity.”*

2. Supplement Section IV Introduction:

- **P5.L2. Added** to address both overfitting and correlated/redundant features that *“When training the estimator, we must ensure that it generalizes to unseen configurations of our processor. In turn, we take significant care in mitigating the risk of overfitting (cited). In traditional statistics and machine learning applications, overfitting is often combated by reducing the complexity of a model*

by discarding or constraining correlated features - e.g. via principal component analysis - or by suppressing them during training - e.g. via regularization. These approaches are most compatible with models that don't necessarily require interpretability, such as neural networks. However, they are incompatible with our requirement that the estimator be physical. Namely, we must keep all physically relevant error components, even though some of them may be correlated in some scenarios. Next we formalize the overfitting problem and then develop training-data sampling and model-training protocols to overcome it."

Response 1.5:

To address "the authors choose not to present these weights, deeming them as not universally beneficial. However, revealing these weights might offer readers deeper insights into the error mechanisms":

While we maintain that the weights are not generally useful, the weighted error components correspond to generally useful error mechanism contributions. We have now exposed these for all 16 configurations explored in the metrology experiment. We also added qualitative and quantitative analyses of these contributions to offer deeper insights into the error mechanisms.

Revisions:

1. **Added new Supplement Section VIII.D. titled "Metrology experiment"**
 - **P16.L71. Added** qualitative and quantitative analyses of the error contributions (i.e. weighted error components) for all 16 metrology experiment configurations.
2. **Added New Supplement Figure S14**
 - **P17. Added** heatmaps with frequencies and errors for all configurations.
3. **Added New Supplement Figure S15**
 - **P18. Added** bar graphs with error contributions for all configurations.
4. **Added new Supplement Figure S16**
 - **P19. Added** quantitative analyses between error mitigation strategies and error mechanisms to quantify their impact and interplay.
5. **Main Section VIII:**
 - **P6.L5. Added** a summary of the new SI section: *"Analyzing error contributions in optimized configurations, we confirm that activating mitigation strategies selectively and effectively suppresses their corresponding error components, while only weakly impacting others (see SI). These results support our interpretation of error components as error mitigation strategies and that our optimizer can effectively reconcile their competition and suppress them."*

2. Optimization's Novelty: The manuscript delves deeply into the workflow and features of the SNAKE optimizer (including snake healing and snake stitching). Yet, much of this information appears to tread on familiar ground. **The optimization segment seems to be a repurposing**

of an established technique in the domain, without proffering any substantial advancements or addressing potential challenges inherent to its application. A statement made by the authors, for instance, reads: "SNAKE is not anticipated to locate globally optimal configurations. Yet, if it identifies configurations that are sufficiently competent at any intermediary scale, it resolves the scaling complexity concern." **It's ambiguous where the boundaries of 'sufficient competence' lie, and whether it signifies a balancing act between performance and intricacy.**

We have split our response into three sections below:

Response 1.6:

To address "The optimization segment seems to be a repurposing of an established technique in the domain, without proffering any substantial advancements":

We ask the reviewer to see **Response 1.1**, which highlights the novelty of this manuscript with respect to our original theoretical proposal (*Klimov, et al. arxiv:2006.04594 (2020)*) and clarifies the significant technical advancements that were necessary to deliver this work.

We also highlight the novelty of the Snake optimizer with respect to established techniques. While Snake leverages some concepts from dynamic programming and graph optimization, it is a novel technique that we developed specifically to meet the flexibility, stability, and scalability demands of an industrial control system. We are not aware of any other optimization system that offers similar features, noting that we have patented them (*Klimov, Patent US11699088B2 (2019)*, <https://patents.google.com/patent/US11699088B2/en>). In addition to the revisions below, we ask the reviewer to see **Response 3.6**, where we addressed a related question.

Revisions:

Expanded Supplement Section V. Introduction to highlight Snake's novelty:

P8.L29. Added that *"Our optimization system is based on the Snake optimizer, which we proposed (cited) as a platform for deploying custom optimization strategies within the demands of an industrial control system. Snake leverages concepts in dynamic programming and graph optimization to offer several key functionalities, which to the best of our knowledge are not offered by any other optimizer:*

- ***Flexibility:*** *Can implement a wide array of optimization strategies via judicious selection of several parameters (see Section V.C.). Most notably, it can deploy virtually any inner loop optimizer at any dimension between the local and global optimization limits. This customizability facilitates adapting Snake to a variety of optimization landscapes.*
- ***Scalability:*** *Runtime scales linearly versus processor size without parallelization and sub-linearly with geometric parallelization ("stitching").*

- **Stability:** *Can locally re-optimize performance outliers (“healing”) as they emerge over time to stabilize processors over long timescales (i.e. months). Healing is much faster than optimization (see Supplementary Section V.E.) and scales sub-linearly with the number of outliers when parallelized.*

In our past proposal, we established Snake's theoretical foundations and software abstractions. In this section, we map frequency optimization into Snake. We then provide broadly applicable implementations of its parameters and employ them to explore tens of optimization strategies between the local and global optimization limits over thousands of simulated optimization threads. Finally, we promote the most promising strategies to the experiments in the main text.”

Response 1.7:

To address “potential challenges inherent to its application”:

Below we outline what we believe to be the most significant challenges inherent to applying our optimization strategy and how we have addressed them.

1. Implementing Snake’s core algorithm.
 - a. This is extensively treated in our original theory proposal (*Klimov, et al. arxiv:2006.04594 (2020) available at <https://arxiv.org/abs/2006.04594>*).
2. Implementing / tuning Snake’s parameters (e.g. traversals, inner-loop optimizer, scope).
 - a. These are extensively treated in this manuscript in **Supplement Section V.C** with concrete implementations, extensions, and benchmarks.
3. Defining the optimization model.
 - a. We took significant steps to add more clarity to this via revisions:

Revisions:

1. Please see **Response 1.2**, **Response 1.3**, and **Response 1.4**.
2. **Supplement Figure S3:**
 - a. **P6. Added** details to facilitate implementing our training procedure.

Response 1.8:

To address “It’s ambiguous where the boundaries of ‘sufficient competence’ lie, and whether it signifies a balancing act between performance and intricacy.”

We agree that this deserves clarifications.

Revisions:

1. Main Section V:

- **P3.L104. Added** that sufficient performance depends on the target quantum algorithm, which for error correction is the fault-tolerance threshold, and added a reference.

2. Main Figure 2b:

- **P4. Added** a “*Best*” label (distinct from the existing “*Default*” label) to better highlight that there is a balancing act involved.
- **P4. Added** to the caption: “*Although S=4 performs best, S=2 offers a better balance between performance and runtime (see SI) and is set as our default*”, referencing the SI where we discuss optimization runtimes at depth.

3. Supplementary Section V.C.:

- **P10.L44. Added** Snake’s parameters must be tuned to the problem of interest “*while balancing runtime and performance, which often compete.*”

3. Metrology Investigation: The discourse on metrology in section VIII seems cursory. The authors elaborate on their methodology: "To discern their impact and comprehend their interactions, we sequentially activate them in varied combinations, optimize, and benchmark CZXEB." Nonetheless, the exposition appears restricted to frequency configurations with singular mitigation strategies in play. Such a methodology yields an oversimplified visual representation, sidestepping a nuanced analysis of the interactions among various mitigation approaches. The concluding remark in this segment, emphasizing the criticality of error metrology on the optimization strategy's performance, feels too generalized and lacks depth.

Response 1.9:

We agree with the Reviewer and took significant steps to add depth to the metrology investigation via qualitative and quantitative analyses as described in **Response 1.5** above.

Reviewer #2 (Remarks to the Author):

The manuscript has a strong contribution to scalable quantum control; I'd recommend acceptance with minor revision. Here I detail my assessments by outlining the most important strengths and weaknesses.

We thank the Reviewer for their time and effort in reviewing our manuscript and for the positive recommendation, pending minor revisions. We are especially thankful for highlighting weaknesses and asking important questions regarding future scalability, including runtime budgets and model-free optimization. We have taken significant steps to address these comments, which we believe have strengthened our message and manuscript.

Strengths:

- * A software approach to scalably optimize the tunable qubit architectures is presented. As hardware size and capabilities grow, manual optimization will have a hard time keeping up. I believe the field will see software advances playing a significant role going beyond the NISQ era. And this work is a timely demonstration.
- * The "snake optimizer" orchestrates the frequency configurations for running algorithms by a divide-and-conquer, model-based optimization strategy. A near linear (to system size) running time scaling is shown while achieving a significant performance advantage.
- * Experimental validation is presented on quantum systems with ~70 qubits. Furthermore, simulated results are also shown to forecast applicability to future systems with ~1000 qubits (for tens of logical qubits).
- * Presented evidence that software-based algorithm-specific optimization is critical.

Weaknesses:

Overall, the manuscript outlines a series of novel software techniques for improving the performance of hardware to match algorithms' demands. However, I believe there are some areas where the software evaluation could be strengthened. I'm optimistic that with the suggested improvements, the manuscript would tell readers a much stronger case about the significance of the snake optimizer.

- * Limited benchmarks. The manuscript presented an extensive analysis of the cross-entropy/randomized benchmarking circuits, but unfortunately **stops short of demonstrating error-correcting operations, such as syndrome measurements or logical operations. The title indicates that the optimization tool flow would contribute to scaling up to logical qubits**, which got me super excited to see how the software could bring us closer to the demonstration of error-corrected logical qubits. **In reality, the software is not tailored to**

error correction, but is general for any applications. I am sure readers would appreciate the work much better if the performance analysis included more benchmark circuits.

We have split our response into two sections below:

Response 2.1:

To address “stops short of demonstrating error-correcting operations, such as syndrome measurements or logical operations. The title indicates that the optimization tool flow would contribute to scaling up to logical qubits”

We agree that understanding the impact of our control optimization strategy on error correction metrics including syndrome measurements and logical operations is critical. A multi-year effort to understand these is under way. However, we strongly believe that the preliminary results of this effort are beyond the scope of this manuscript and warrant dedicated future publications. Nonetheless, we have clarified how our paper is geared towards error correction and how it has already contributed to scaling up to logical qubits.

Revisions:

- 1. Main Section II.:**
 - a. P2.L16. Added** that “*we configure the processor to execute the surface code gate set*”.
- 2. Main Section III.:**
 - a. P2.L36. Added** that “*CZXEB reflects the structure of the surface code’s parity checks and has empirically served as a valuable performance proxy of logical error (cited)*”.
- 3. Main Section IV.:**
 - a. P3.L25. Added** that we set our target quantum algorithm to “*CZXEB to gear the estimator towards the surface code’s parity checks*”.
- 4. Main Section XI.:**
 - a. P7.L47. Added** that “*A recent demonstration of error suppression in a scaled up surface code logical qubit (cited) enabled by this strategy underscores its potential*”.

Response 2.2:

To address “the software is not tailored to error correction, but is general for any applications. I am sure readers would appreciate the work much better if the performance analysis included more benchmark circuits.”

We agree with the Reviewer that our optimization strategy is general and applies to applications beyond error correction. Although we believe that adding more benchmark circuits goes beyond

the scope of this study, we highlight that some elements of our strategy have been previously applied (details previously not disclosed) to optimize other quantum operations and algorithms.

Revisions:

1. Main Section XI.:

- a. **P7.L50. Added** that “*Elements of our strategy have also been employed to optimize quantum operations including measurement (cited) and SWAP gates (cited), and quantum algorithms for optimization (cited), metrology (cited), simulation (cited), and beyond classical computation (cited).*”

* Limited software details. The manuscripts omitted some details of the software (e.g., in appendices), **making the software hard to reproduce**. I suspect this is due to intellectual property considerations, as the software is not open-sourced. For this reason, as a reviewer, I cannot check the functionality of the tool or the reproducibility of the results. As a result, my assessment is primarily focusing on evaluation methodology and results.

Response 2.3:

The Reviewer is correct that our software cannot be open-sourced due to intellectual property considerations. However, the core Snake optimization algorithm should be reproducible via our past theoretical proposal, which is cited (*Klimov, et al. arxiv:2006.04594 (2020) available at <https://arxiv.org/abs/2006.04594>*). Furthermore, we have taken significant steps to add details throughout the manuscript to facilitate reproducibility of novel technology that has not been previously reported. For example, please see **Response 1.7.**

* Some clarification questions:

1. The optimization relies on an accurate "algorithm error estimator". Can the authors elaborate on why is a linear model (across different error mechanisms) sufficient?

Response 2.4:

We agree with the Reviewer that our motivations for using a linear model should be clarified. We also note that we addressed a similar question in **Response 1.2.**

Revisions:

1. Supplement Section III Introduction:

- **P2.L29. Expanded** that “*The algorithm error estimator can be represented by a variety of models (e.g. a linear model, a neural network, or a quantum simulation), ...*”

2. Supplement Section III.A.:

- **P2.L57. Added** the rationale behind each assumption underlying the noise model, including that a linear model over error mechanisms “*should apply in the limit of small and uncorrelated error components*”, which we believe we are in.
- **P3.L8. Added** that we have focused on a linear model since it “*establishes a powerful and broadly applicable framework*:
 - *Compatible with physicality, speed, and accuracy.*
 - *Error components can be added as new error mechanisms are discovered.*
 - *Error components can be interchanged to transfer the estimator between hardware architectures (**reference to new Supplement Section IIID**).*
 - *Weights can be re-trained to transfer the estimator between quantum algorithms.*
 - *Optimization variables (F here) can be interchanged to adapt the estimator to other control variables, provided that the error components are defined in terms of those variables.*
 - *Equivalent to the basis-expansion method in machine learning (cited), bridging the domains and facilitating knowledge transfer.*
- **P3.L27. Added** that when a quantum algorithm does not fit within our assumptions, “*methods like randomized compiling (cited) offer the potential to recompile those algorithms into compliance*”.

2. The running time budget of the estimator and subsequently the overall tool flow is determined based on the impact of system drifts and outlier probability. How does this budget change as hardware evolves? Can we forecast quantitatively how fast the optimization needs to be for error correction?

Response 2.5:

The reviewer is correct on how our runtime budget is informed. We added additional information on how the budget relates to error correction and our expectations as the hardware evolves.

Revisions:

1. Main Section X.:

- **P7.L13. Added** that our runtime budget “*was chosen for compatibility with large surface codes*” and referenced the **new Supplement Section V.D.**

2. Added new Supplement Section V.D. titled “Optimization runtime budget”.

- **P12.L33. Added** that “*We develop a runtime budget with the objective of operating a distance 23 surface code logical qubit with $N=1057$ qubits. Since the surface code has a lenient qubit failure tolerance of $>1\%$ (cited) and since our outlier emergence probability is ~ 0.01 / 24 hours / gate (see Section VIII.B), we have ~ 24 hours to characterize, optimize, calibrate, and benchmark our processor and finally execute the algorithm before restarting the process.*”

If we target 12 hours for executing the algorithm, we have 12 hours to go from characterization to benchmarking. Since characterization, calibration, and benchmarking take ~2 hours and are nominally independent of processor size due to parallelization, we have ~10 hours left. From this large window, we only budget 0.5 hours for optimization, leaving ~9.5 hours for unforeseen scaling overhead.

The 0.5 hour runtime budget can be fulfilled via a combination of optimization, healing, and stitching (see Section V.E). Furthermore, as the hardware evolves, we expect the outlier emergence probability to decrease and the control system to become faster through hardware and software advancements, which should relax all budgets and enable operating even larger surface codes."

3. Improved Supplement Figure S8

- **P12. Expanded** into distinct panels for "optimization", "healing", and "stitching" runtimes, with points of interest to support that our runtime budget is compatible with operating large surface codes.

3. Is the ellipsis in SI, section II there to indicate some parameters are omitted? For what reasons? Similarly, for the error components in SI, section V.B.

Response 2.6:

We agree that the ellipses should be addressed.

Revisions:

1. Supplement Section II.:

- **P2.L24. Removed** the ellipses, which intended to mean that there are other characterization data, which are not relevant to our work. We believe this is already clear from the surrounding context.

2. Supplement Section III.B.:

- **P3.L43. Added** that we provide "key" error components and removed the ellipses, which intended to mean that there is flexibility in the error components depending on the problem. We believe this is already clear from the surrounding context, including our new **Supplementary Section III.D.** with extensions of this framework to other hardware.

4. There is a claim in "Outlook" that the optimizer can deploy model-free RL agents. Do we expect a crossover point where model-based optimization will no longer be feasible and model-free models will be required? At the moment, it is hard to see how the model-based optimization techniques proposed in this work will persist and carry over to scale beyond thousands of qubits. With that said, I completely agree with the authors that there's value in fleshing out what we can do at the 100-1000 qubit scale first.

Response 2.7:

We are optimistic that we can brute-force scale our strategy to tens of thousands of qubits through stitching and parallelization of classical resources. However, at those scales, hardware inhomogeneities, control inhomogeneities, and/or other unknowns may degrade its performance advantage. We believe that model-free agents could help overcome these challenges and are excited to explore this research frontier.

1. Main Section XI.:

- **P7.L74. Added** that model-based optimization should remain critical for injecting metrology into control optimization “*for the foreseeable future*” and referenced the SI, where we expand as described next:

2. Supplement Section V.C.4.:

- **P12.L24. Added** that “*In the short term, model-free agents could refine configurations found via model-based optimization to compensate for inaccuracies in the algorithm error estimator, which are expected to increase with processor size due to increased control and hardware inhomogeneities. In the longer term, they could replace model-based optimization entirely (cited) and eliminate the research burden of developing performance estimators.*”

Reviewer #3 (Remarks to the Author):

Dear Editor,

I have read the work "Optimizing quantum gates towards the scale of logical qubits", by Klimov et al., which the Google Quantum AI team present a classical strategy to optimize frequency controls of all qubits in a quantum processor, with the goal of minimizing the expected error per gate.

The work makes various interesting points, in my opinion. One is the algorithm itself, which is a greedy optimization method to select frequency schedules for qubits, so as to minimize the total expected error. The second one, equally important in my opinion, is the formulation of the cost function itself. This is a nontrivial step, because an inadequate error model leads to an inadequate optimization. Finally, the work itself is a tour-de-force calibration of a very large quantum processor to very good accuracy, for current standards, which makes this work a state-of-the-art reference ---absent from other large quantum computer providers--- and demonstrates the utility of the method.

However, there are presentation problems in the material, which is quite dense and at points does not offer information which is critical to understand the algorithm and its performance. Therefore, I would like the authors to address my comments below before committing to a final acceptance decision.

We thank the Reviewer for their time and effort in reviewing our manuscript and for the perspective that this work represents a "state-of-the-art reference". We found the detailed technical comments to be extremely useful and used them as a guide to improve our manuscript.

First of all, I think that error model information should not be completely relegated to the supplementary material, because it can be included with different wordings or a few sentences:

1. Section IV omits important facts regarding the error model. The first one, which was not obvious to me (and is still not fully clear) is that the error model **only cares about which gates are present in the algorithm** and **not the order of execution or repetition**.

Response 3.1:

We agree that this deserves clarifications. We have clarified that both the *local* temporal order of execution and repetitions are included in the estimator. We also clarified the assumptions under which an additive error model - which discards long-range temporal order - is justified.

Revisions:

1. Main Section IV.:

- **P3.L32. Added** that our error components consider error mechanisms “*over qubit frequency trajectories*” and referenced the SI for additional details.
- 2. Supplement Section III.A.1.:**
- **P2.L45. Added** that the error model includes repetitions via “*(i.e. CZXEB with 2 qubits and m cycles comprises 2m SQ and m CZ gate error estimators).*” We also note to the Reviewer that - depending on the algorithm - symmetries can be exploited. For example, for CZXEB, only one cycle needs to be considered.
 - **P2.L48. Added** that an additive error model, which discards long-range temporal order, is “*motivated by the digital error model ...*”.
- 3. Supplement Section III.B.:**
- **P4.L19. Added** that “*Two-qubit error components typically integrate errors over frequency trajectories F according to the hardware implementation and the local temporal order of gates within the quantum algorithm. For CZXEB, they may integrate over a trapezoidal trajectory that links idles and interactions*”

2. After various readings of the manuscript and supplementary material, I still don't understand what structure is followed by the weights " $w_{\{g,m\}}$ " and how they are calibrated.

Response 3.2:

We thank the Reviewer for bringing to our attention that more information on the weights and how they are trained would be useful.

Revisions:

- 1. Main Section IV.:**
- **P3.L38. Added** that “*we train the weights via a protocol that we developed specifically to reduce the risk of overfitting (see SI). It constrains weights via homogeneity and symmetry assumptions and then leverages the frequency tunability of our architecture to train them} on single- and two-qubit gate benchmarks taken in configurations of variable complexity.*”
- 2. Supplement Section III.A.3.:**
- **P2.L70. Added** that “*One unique weight is assigned to, and multiplies, each error component*” to help clarify their structure.
- 3. Supplement Figure S3:**
- **P6. Added** details to clarify the structure of the weights and how they are trained.
- 4. Supplement Section IV. Introduction:**
- **P5.L2. Added** that “*When training the estimator, we must ensure that it generalizes to unseen configurations of our processor. In turn, we take significant care in mitigating the risk of overfitting (cited). In traditional statistics and machine learning applications, overfitting is often combated by reducing the complexity of a model by discarding or constraining correlated features - e.g. via principal component analysis - or by suppressing them during training - e.g. via regularization. These approaches are most compatible with models that don't*

necessarily require interpretability, such as neural networks. However, they are incompatible with our requirement that the estimator be physical. Namely, we must keep all physically relevant error components, even though some of them are correlated in some scenarios. Next we formalize the overfitting problem and then develop training-data sampling and model-training protocols to overcome it.”

3. The third important fact which is absent from Sect. IV is that the formula at the end of page 2 assumes homogeneous parameters. This was implicit in the sentence "has 16 trainable parameters", but was unclear to me -- I originally thought 16 trainable parameters per error component.

Response 3.3:

Please see our Response 3.2 for more information on the structure of the weights and how we calibrate / train them. Additionally, we have clarified:

Revisions:

1. Main Section IV.:

- a. **P3.L39. Added** we constrain weights “*via homogeneity and symmetry assumptions*”
- b. **P3.L47. Added** that the estimator has 16 trainable weights “*for the full processor*”.

4. The second paragraph on page 3 enumerates the error models. That list seems shorter than the one in the supplementary, which ends with some dots "... (?!?). It would not hurt splitting the number 4×10^4 as a product of error models times number of subsets, or a more informative list that is consistent with the procedure. This decomposition is also incomplete in the supplementary material.

Response 3.4:

We thank the reviewer for highlighting that the large number of error components could use more transparency.

Revisions:

1. Supplement Section III.B.:

- **P3.L43. Added** that we provide “*key*” error components and removed the ellipses, which intended to mean that there is flexibility in the error components depending on the problem. We believe this is already clear from the surrounding context, including our new **Supplementary Section III.D.** which describes extensions of this framework to other hardware.

- **P3.L46. Added** To clarify why the list in the Supplement is longer than in the Main text: *“We note that each error mechanism is segmented into multiple error components as described in detail below. When referencing a mechanism, we consider all such components together.”*
- **P3.L56. Added** a “Scale” item for each error component that exposes how many such components there are for an N qubit processor, once segmented.
- **P4.L12. Added** that *“an N qubit processor has $\sim 10^3 N$ error components.”*
- **P4.L14. Added** additional segmentation details after *“This number is large due to the high granularity with which we segment error components ...”*
- **P3.L50. Reordered** the error component lists to start with mechanism names (e.g. “dephasing”) to facilitate connections with the main text. Also used a shorter notation for frequencies for clarity.

I also have a fundamental issue with the choice of the words “frequency trajectory choreography”, because an essential component of a choreography is the order in which steps and poses are executed. This ingredient seems to be absent in the error model and the optimization itself, unless I am mistaken, which is why I had such a difficult time understanding the formulation. If what I say is correct, the authors should correct this clearly in the abstract and the introductory text.

Response 3.5:

We ask the Reviewer to see our Response 3.1 above. To summarize, we typically integrate two-qubit error components over qubit frequency trajectories (i.e. from idles to interactions and back to idles). In turn, this embeds the local time-ordering of gates in CZXE (i.e. SQ then CZ then SQ). Therefore, we believe the phrase “frequency trajectory choreography” is justified.

There are also presentation issues with the optimizer:

1. The optimization algorithm, while clever, **seems like a standard greedy optimization method, which is the first thing one would try**. This method seems to be explained in an e-print from 2006 (<https://arxiv.org/pdf/2006.04594.pdf>), which the authors properly cite, but it is unclear to me whether there has been any evolution. The original manuscript already contained the notions of scope, greedy search and stitching, but lacked a proper description of the error model / cost function and had no experimental data, which are the two big assets in this contribution. **The authors should more clearly describe whether the optimization strategy is identical or new ingredients have been explored.**

Response 3.6:

We ask the Reviewer to see Response 1.1 and Response 1.6, where we addressed how this work differs from our original theoretical proposal for Snake, which the Reviewer referenced (*Klimov, et al. arxiv:2006.04594 (2020)*, though we note it is from 2020 and not 2006 as implied

by the arXiv label.), and the significant technical advancements were necessary to deliver this work. Additionally, we respond directly to the reviewer's questions below:

To address “seems like a standard greedy optimization method, which is the first thing one would try” and “should more clearly describe whether the optimization strategy is identical or new ingredients have been explored.”

We agree with the Reviewer that the first things that one would likely try are optimizing at the greedy-local or non-greedy-global limits. One of Snake's distinguishing features is that it can implement a wide array of optimization strategies *at either of these limits or anything in between*, by tuning the scope parameter. We are not aware of any other optimizer that offers this flexibility, noting that we have patented the underlying methods (*Klimov, Patent US11699088B2 (2019)*, <https://patents.google.com/patent/US11699088B2/en>).

This flexibility enabled us to explore a wide array of optimization strategies between the greedy local and non-greedy global limits in both simulation (>30 strategies at 7 greediness levels benchmarked in **Supplementary Figure S7**) and experiment (the best simulated strategies at each of the 7 greediness level benchmarked in **Main Figure 2b**).

Interestingly, we found that neither the greedy local nor the non-greedy global optimization limits are performant. Instead, we found that intermediate dimensional optimization, enabled by Snake, was most performant. Furthermore, we found that the most performant configurations were found *towards* the greedy limit, strongly supporting the scalability of our approach.

Additionally, we note that we expect the optimal greediness / scope to depend strongly on the hardware architecture. The fact that greediness (i.e. scope) is one of Snake's tunable parameters makes it especially broadly applicable and attractive for a research environment with rapidly evolving hardware (more in **Response 1.3**).

Revisions:

1. Supplement Section V. Introduction:

- **P8.L36. Expanded** to highlight Snake's novelties with respect to other optimizers, including that it *“Can implement a wide array of optimization strategies via judicious selection of several parameters (see Supplement Section V.C.). Most notably, it can deploy virtually any inner loop optimizer at any dimension between the local and global optimization limits. This customizability facilitates adapting Snake to a variety of optimization landscapes.”*
- **P8.L54. Added** that we *“provide broadly applicable implementations of [Snake's] parameters and employ them to explore tens of optimization strategies between the local and global optimization limits over thousands of simulated optimization threads. Finally, we promote the most promising strategies to the experiments in the main text.”*

2. Supplement Section V.C.:

- **P10.L41. Added** that “*Snake can implement a wide array of optimization strategies through judicious selection of four parameters ...*”.

3. Main Figure 2b:

- **P4. Added** to the caption: “*intermediate dimensional optimization ($2 \leq S \leq 4$) outperforms both local and global optimization*”.

2. The authors should declare, even if tentatively or on average, the number of values "k" that frequencies can take. It is unclear from the supplementary material, whether the figure of 2^{1437} corresponds to the total configurations for one qubit, or the total processor.

Response 3.7:

We agree this could use more clarity.

Revisions:

1. Supplement Section V.B.:

- **P10.L28. Added** “*From these operating bandwidths, together with a 2 MHz hardware discretization, we can estimate the average number of idle and interaction frequency options (k in Section V of the main text) and the total number of frequency configurations for the processor*”
- **P10.L33. Added** “*Idle frequency options*” and “*Interaction frequency options*”.
- **P10.L36. Added** that 2^{1437} corresponds to the total configurations “*for our processor*”.

3. Regarding the previous point, I am confused by the fact that the authors suggest a discretized frequency variable, but the Snake algorithm seems to rely on continuous optimization locally, according to III.3 in the supplementary material, from their mention of L-BFGS and similar algorithms.

Response 3.8:

We agree that this is an important point. In addressing it, we have hopefully further highlighted the flexibility in Snake’s inner-loop optimizer and the optimization strategies that it can deploy.

Revisions:

1. Supplement Section V.C.4.:

- a. **P12.L20. Added** that Snake can deploy “*continuous or discrete inner-loop optimizers, treating the optimization variables accordingly*”

4. It was also unclear to me whether the greedy optimizer goes over optimized values more than once. On a second read, it does not seem so, but this is not to be discarded in this type of algorithms. Some words on this would be helpful.

Response 3.9:

The Reviewer raises an excellent point. We have indeed seen that multiple optimization iterations can boost performance. This more advanced strategy was not benchmarked in our manuscript to not confound it with the performance of the core strategy - which as we demonstrated is sufficient to approach the error correction threshold - and to make the work more generally accessible. Nonetheless, we have added this extension to the supplement.

Revisions:

1. Supplement Section V.C.2:

- a. **P11.L19. Added** that *“Multiple optimization traversals, which we consider an extension of healing, may also boost performance”*.

Other questions:

- The lack of finer-grained parallelization is attributed to the problem with stitching configurations. Wouldn't that be solved by allowing the healing process to run over optimized qubits?

Response 3.10:

The Reviewer raises an important point. The limits of parallelization are not well understood and will be experimentally investigated on larger processors once available.

Revisions:

1. Main Section X.:

- **P7.L27. Added** that *“We chose convenient stitch geometries, but believe they will ultimately need to be optimized”* and added a new supplement section:

2. Added new Supplement Section VIII.F. titled “Stitching experiment”:

- **P20.L17. Added** *“The stitching demonstrations in Figure 4 employed convenient stitch geometries. Even though stitched-and-healed configurations performed as well as their unstitched counterparts, we expect that the number of stitched regions and seam geometry will ultimately need to be optimized. First, we expect that progressively increasing the number of stitched regions - which would favorably lead to shorter optimization runtimes - will eventually start to degrade performance as more constraints between more independently optimized configurations will have to be reconciled. Second, we expect that optimizing the seam geometry will be necessary for applications like error correction, where the geometry of poorly performing gates is particularly important (cited)”*

- I find the analysis of the different error contributions quite interesting, though it is misleading the use of the word "error mitigation strategy", because the error mitigation may (or may not) result from that particular cost function. In this regard, I find the surface plot associated to pulse distortion quite puzzling, in that the optimization should find a uniform landscape, instead of strongly biasing two-qubit gates on the south-east boundaries. Given the homogeneous error parameters, this result and other inhomogeneities in the other surface plots merit discussion.

Response 3.11:

We agree with the Reviewer that this is important. We respond below and note that we addressed a similar question in Response 1.5.

Revisions:

- 1. Added new Supplement Section VIII.D., new Figure S14, new Figure S15, and new Figure S16.**
 - **P16.L71. Expanded** our qualitative and quantitative analysis of the metrology experiment, which supports our association of error components with mitigation strategies.
- 2. Main Section VIII.:**
 - **P6.L5. Added** that *"Analyzing error contributions in optimized configurations, we confirm that activating mitigation strategies selectively and effectively suppresses their corresponding error components, while only weakly impacting others (see SI). These results support our interpretation of error components as error mitigation strategies and that our optimizer can effectively reconcile their competition and suppress them."*
 - **P5.L36. Added** that the surface plots are not uniform due to *"inhomogeneities arising from fabrication imperfections in the processor's parameters"*.
 - **P5.L45. Added** that *"pulse-distortion mitigation biases idles towards a multi-layered checkerboard, with neighbors at one of two symmetric $|11\rangle \leftrightarrow |02\rangle$ CZ resonances (cited), and interactions towards resonance between the idles, to minimize frequency excursions."*
 - **P5.L49. Added** that *"The inversion of frequencies at the eastern edges of the processor was triggered by fabrication imperfections that broke the symmetry between CZ resonances. This observation highlights non-trivial interplay between error mitigation and hardware inhomogeneities."*

Additional Revisions

In addition to addressing Reviewers' comments, we made the following non-essential revisions:

1. Wording changes throughout the **Main** and **Supplement** for clarity and flow.
2. Vectorized all figures in the **Main** and **Supplement**.
3. Reordered several **Supplementary Sections** for clarity and flow, notably:
 - a. Split “**The Algorithm Error Estimator**” into “**The Algorithm Error Estimator**” and “**Training the Estimator**” and moved them before “**Optimization System**” for better correspondence with the **Main** text.
 - b. Moved “**Optimization runtime scalability**” to a subsection of “**Optimization System**”.
 - c. Moved “**Algorithm specificity**” to a subsection of “**Additional Experimental Details**”.
4. **P9. Added** to **Figure S6** 68% confidence intervals for CZXEB benchmarks.
5. **P15. Combined** in **Figure S13** the two figures that were in **Supplement Section VII. titled “Simulation Environment”** to clarify the connection between them.
6. **P20. Added** to **Figure S17** SQRB data for completeness.

REVIEWERS' COMMENTS

Reviewer #1 (Remarks to the Author):

The authors have satisfactorily addressed all of my questions, and the revised manuscript is now in excellent condition. However, I maintain that the scientific novelty of this work is insufficient, despite the impressive and extensive efforts made, particularly in the engineering aspect. I defer the final decision to the editor.

Reviewer #2 (Remarks to the Author):

I would like to thank the authors for answering and addressing my comments in their response. I have read the response letter and the revised manuscript. I recommend acceptance.

The revised manuscript has sufficiently addressed my concerns over scalability and runtime budget. I see a significant value in this work as it informs the community about the importance of software optimization for quantum control. It is pity that such a tool is closed-sourced and not available to use by the general research community. So, my concern over software reproducibility remains. I do believe the scientific value of the work would multiply if (some version of) the tool can be used by the broader community in the future.

Reviewer #3 (Remarks to the Author):

I have read with great pleasure both the revised manuscript and the replies to my objections. To be clear, the most relevant ones regarded the details of the error model and calibration, and the nonincremental nature with respect to the original Snake algorithm. I am satisfied with both discussions, as well as with the replies to all of the comments. I find the manuscript is more accessible, specially with respects to the improved supplementary material.